# Living with Restrictions. The Perspective of Nursing Students with Primary Dysmenorrhea

**DOI:** 10.3390/ijerph17228527

**Published:** 2020-11-17

**Authors:** Elia Fernández-Martínez, Ana Abreu-Sánchez, Juan Francisco Velarde-García, María Teresa Iglesias-López, Jorge Pérez-Corrales, Domingo Palacios-Ceña

**Affiliations:** 1Department of Nursing, University of Huelva, Avenida Tres de Marzo s/n, 21071 Huelva, Spain; elia.fernandez@denf.uhu.es (E.F.-M.); abreu@denf.uhu.es (A.A.-S.); 2Department of Nursing, Red Cross College, Instituto de Investigación Sanitaria Gregorio Marañón (IiSGM), Universidad Autónoma de Madrid, Avenida Reina Victoria 28, 28003 Madrid, Spain; 3Faculty of Health Sciences, Universidad Francisco de Vitoria, Crta. Pozuelo—Majadahonda km 1800, 28223 Pozuelo de Alarcón, Spain; m.iglesias.prof@ufv.es; 4Department of Physical Therapy, Occupational Therapy, Physical Medicine and Rehabilitation, Research Group of Humanities and Qualitative Research in Health Science of Universidad Rey Juan Carlos (Hum & QRinHS), Avenida Atenas s/n, 28922 Alcorcón, Spain; jorge.perez@urjc.es (J.P.-C.); domingo.palacios@urjc.es (D.P.-C.)

**Keywords:** dysmenorrhea, pelvic pain, nursing students, qualitative research

## Abstract

Primary dysmenorrhea (PD) affects a large number of female university students, diminishing their quality of life and hindering academic performance, representing a significant cause of absenteeism. The purpose of our study was to determine how nursing students experienced restrictions as a result of primary dysmenorrhea. A qualitative exploratory study was conducted among 33 nursing students with primary dysmenorrhea. A purposeful sampling strategy was applied. Data were collected from five focus groups (two sessions each) and the field notes of 10 researchers. A video meeting platform was used to conduct the focus groups. A thematic inductive analysis was performed. Thirty-three female nursing students participated in the study with a mean age of 22.72 (SD 3.46) years. Three broad themes emerged: (a) restrictions on daily activities and sports; (b) academic restrictions, and (c) restrictions on social and sexual relationships. The students described restrictions in performing everyday activities, such as carrying weight, and shopping. Some students even gave up the practice of sports and were absent from classes at the university, and from clinical practices at the hospital. The pain affected their ability to maintain and create new social relationships. Primary dysmenorrhea caused restrictions in the personal, social and academic life of the nursing students.

## 1. Introduction

Menstrual pain is a very common problem among young women, estimated to afflict between 67 and 74% of young women worldwide [1]. In Spain, up to 75% of female university students suffer from this condition [2]. Dysmenorrhea is defined as chronic spasmodic pelvic pain associated with menstruation and often accompanied by other symptoms such as nausea, vomiting, diarrhoea, dizziness, headache, irritability, and depressive symptoms [3]. Two types of dysmenorrhea are distinguished in the literature: primary dysmenorrhea (PD) and secondary dysmenorrhea (SD) [4,5]. PD is not caused by a pelvic pathology and is principally thought to be due to an excess of prostaglandins although it may also be associated with dysfunctions in the immune, endocrine and vascular systems [6]. This is the most common type of dysmenorrhea, affecting an estimated 63% to 75% of Spanish female university students [7,8]. In contrast, SD is associated with other diagnosed pathologies, most frequently endometriosis [9].

The principal therapeutic treatment for both types of dysmenorrhea is non-steroidal analgesics (NSAIDs), followed by anovulatory drugs [6]. Previous studies show that some non-pharmacological methods, such as exercise and local heat are effective in preventing and alleviating pain [10,11]. However, certain studies point to self-medication as a habitual strategy of pain management without consulting a healthcare professional [12,13].

Iacovides et al. and Al Jefout et al. [14,15] reported that dysmenorrhea can significantly decrease the quality of life of women. Previous studies have also shown [1,2,16,17,18] that dysmenorrhea can negatively impact academic performance. Moreover, it has been related to decreased concentration, among other symptoms, as well as greater absenteeism during menstruation and presenteeism, understood as the practice of attending classes or clinical placements when suffering from severe menstrual pain.

At present, there are few qualitative studies [19,20,21,22] that describe and analyse the perspective and experience of young female students suffering from dysmenorrhea and how it influences their daily lives. Previous studies [19,20,21,23] have reported how dysmenorrhea limits women’s daily life. Chen et al. [19] reported that the effects of dysmenorrhea can include difficulty in sitting and walking, the need to adopt antalgic positions (sleeping in foetal position), absenteeism from university classes and/or work, abandonment of family responsibilities (accompanying children to school) and recreational activities. A study by Abreu-Sánchez et al. [2] showed how dysmenorrhea interferes with the academic life of university students, to the point of being absent from class or clinical placements. The study also noted that women feel misunderstood and worry about the repercussion that dysmenorrhea may have on their academic performance. The experience of these women needs to be identified and analysed to facilitate the establishment of comprehensive healthcare programs to address their needs. The purpose of this study was to identify the restrictions experienced by Spanish nursing students in relation to PD. The question guiding our study was: what limitations and/or restrictions affect nursing students with dysmenorrhea?

## 2. Materials and Methods

### 2.1. Design

A qualitative exploratory study was conducted using focus groups (FGs) [24]. Qualitative methods and FGs are helpful for understanding the experiences and motivations of participants, identifying unexplored situations, confirming hypotheses, creating questionnaires and designing intervention programs [25,26]. Previous studies have used qualitative research techniques and FGs to study chronic pelvic pain and dysmenorrhea [27,28].

### 2.2. Research Team

Six researchers (three men and three women) participated in this study, three of whom had experience in qualitative studies (D.P.-C., J.P.-C., J.F.V.-G.). All researchers had PhDs in health sciences and were not involved in clinical activity. Prior to the study, the positioning of the researchers was established in terms of their theoretical framework, beliefs, prior experiences and their motivations for conducting this research [29,30].

### 2.3. Sampling Strategies and Participants

Purposeful sampling was carried out, selecting participants for their capacity to provide relevant information in response to the research questions [26]. The inclusion criteria consisted of: (a) nursing students enrolled in the University of Huelva (https://enfe.acentoweb.com/) during the study period, (b) PD: who acknowledged experiencing menstrual pain for which no underlying cause had been diagnosed [7,31], (c) suffering from menstrual pain at least once in the last six months, with at least three periods per year [7,8,9,10,11,12,13,14,15,16,17,18,19], (d) moderate-severe pain intensity based on the visual analogue scale (VAS equal to or greater than 4 out of 10 [8,32,33], (e) for at least the three cycles prior to the study [34,35], (f) with normal menstrual characteristics in terms of cycle length, duration of menstruation, quantity and regularity. Normal menstrual characteristics were considered to be periods occurring every 24 to 38 days, on a regular basis, with bleeding that lasts 4.5 to 8 days, and 5 to 80 mL blood loss per cycle [36,37,38], and (g) not being a candidate for requesting diagnostic tests due to suspicion of suffering secondary dysmenorrhea or any other diagnosed pelvic or gynaecological problem based on the recommendations of the Primary Dysmenorrhea Consensus Guideline of the Society of Obstetricians and Gynaecologists of Canada and the Committee Opinion on Adolescent Health Care Dysmenorrhea and Endometriosis in the Adolescent developed by the American College of Obstetricians and Gynecologists [5,34,39].

The exclusion criteria consisted of: being diagnosed with secondary dysmenorrhea (endometriosis, adenomyosis, uterine myomas, cervical stenosis and obstructive lesions of the genital tract) or any other diagnosed pelvic or gynaecological problem (pelvic inflammatory disease, pelvic adhesions, irritable bowel syndrome, inflammatory bowel disease, interstitial cystitis, mood disorders and myofascial pain) or menstrual characteristics that may raise suspicion of these pathologies [5,39], and unwillingness to participate in the study. Thirty-three female nursing students participated in the study. There were no dropouts.

### 2.4. Data Collection

To evaluate different perspectives, FGs were held to acquire an understanding of the problems faced by the group [26]. The FGs comprised female nursing students enrolled at the University of Huelva. The students were randomly assigned to the FGs to avoid any selection bias among the nursing student groups. Five FGs were formed of the students willing to participate in the study. Each FG consisted of 6 to 9 participants, considering that meaningful discussions can be difficult to sustain in groups smaller than four people, whereas groups larger than 10 can prove difficult to manage [40]. The recordings lasted for a total duration of 522 min (290 min in FG session 1, and 232 min in FG session 2), with a mean duration of 52.2 min (58 min in FG session 1, and 46.4 min in FG session 2). See Table 1, below:

### 2.5. Focus Group Procedure

The FGs were conducted using the Zoom digital platform. This platform enabled the simultaneous participation of the groups after receiving an invitation by the research team with audio and/or video recording [41]. Each participant had the option to participate in the FGs with or without activating their video camera, to be recorded in audio or video. At the beginning of the FGs, each participant identified themselves upon entering the platform, so that all participants could be recognised while participating.

The FGs were conducted by a moderator and an observer. The moderator posed questions to which each participant responded, speaking in turns. To request participation, participants raised their hands in the application chat box and the moderator assigned the order of participation. Thereafter, the moderator posed further questions, based on the issues that were brought up in the discussion, in order to further explore or clarify aspects, either individually or with the whole group [30]. The observer supported the moderator, identifying key points and taking notes. To report on the participants’ gestural communication and/or new areas of interest to be investigated, the observer privately messaged the moderator via the corresponding chat box. The FGs were conducted in Spanish. A question guide was used, which was focused enough to gather information on the area of study, although open enough to stimulate discussion and interaction among the participants [26,30]. See Table 2. Nonetheless, data collection in qualitative studies is flexible, allowing the researcher to ask open-ended questions on topics that are relevant to the participants in relation to the research question [26]. Consequently, during the FGs, the moderator asked about those areas of interest that the participants raised in relation to the research question.

All the FGs were audio-video recorded with the prior permission of the participants. Data collection continued until the researcher achieved information redundancy, at which point no new information emerged from data analysis. This occurred in FG5 [26].

### 2.6. Data Analysis

Data collection was based on a full verbatim transcript of the FGs and the researchers’ field notes [26,42]. A thematic analysis [42] was made of the data, with an initial descriptive analysis of the transcribed texts (words, sentences and metaphors directly from the text). The data were then reduced to codes in order to identify emerging topics. These codes were clustered into categories to define the main topics. The final outcome was the identification of themes emerging from the data. No qualitative software was used on the data. Three researchers with experience in qualitative studies (J.F.V.-G., J.P.-C., D.P.-C.) performed the analysis of the FG data. First, an analysis of each FG was performed. Afterwards, the results of the initial analysis were subsequently merged in joint sessions, during which the data collection and analysis procedures were discussed. In the case of differences of opinion, theme identification was decided by consensus.

### 2.7. Ethics and Quality Criteria

This study was approved by the Research Ethics Committee of Huelva (Code: 9/19) and conducted in accordance with the Declaration of Helsinki. All participants provided written informed consent prior to participating in this study.

The study adhered to the guidelines for conducting qualitative studies established by the Standards for Reporting Qualitative Research [43] and the Consolidated Criteria for Reporting Qualitative Research [29]. For trustworthiness, researcher triangulation and participant validation were used to control the credibility of the data. Transferability was established by providing details of the researchers, participants, sampling strategies, data collection and analysis procedures. For dependability, an audit was conducted by an external researcher. Finally, confirmability was carried out by applying researcher reflexivity [25].

## 3. Results

Thirty-three female nursing students participated in the study. The mean age of the participants was 22.72 (SD 3.46) years and the BMI was 21.52 (SD 2.50) kg/m^2^. Up to 87.9% lived in an urban setting compared to 12.1% in a rural setting. A total of 84.8% regularly took some type of medication to manage their menstrual pain, whereas 81.8% used non-steroidal anti-inflammatory drugs and 21.2% used hormonal contraceptives. The study duration was from 1 January 2020 to 1 June 2020. Three broad themes emerged: (a) restrictions on daily activities and sports; (b) academic restrictions; and (c) restrictions on social and sexual relationships. To facilitate the traceability and identification of the results, these are accompanied below by excerpts of transcripts. 

Dysmenorrhea is experienced as an obstacle, restricting the life of the students in three principal dimensions: (a) restrictions on daily activities and sports; (b) academic restrictions; and (c) restrictions on social and sexual relationships.

### 3.1. Theme 1. Restrictions on Daily Activities and Sports

The students reported that PD caused discomfort and pain that was periodic or continuous, which limited their ability to carry out their daily tasks and sports activities.

The pain was neither predictable nor controllable, appearing suddenly and limiting normal activities: “*it is a pretty intense type of pain and it limits normal activities of daily life on many occasions*” (FG 4). Normal activities are inhibited due to pain appearing when making any minimal physical exertion, such as carrying the shopping, lifting a weight or moving furniture, climbing stairs, driving or sitting. It can even hinder the ability to care for one’s children: “*the days when I have pain I can’t take my daughter in my arms; I see stars. Any task, like climbing the stairs at home, getting into the car, going shopping I can’t move*” (FG 4). When the pain appears, the students report that they are forced to interrupt their normal life or any activities and lie down, along with taking medication to manage the pain and/or be able to continue functioning normally: “*It makes doing anything impossible. When it starts, I have to go to the sofa, or lie on the bed, unless I take some analgesic to continue being able to function normally*” (FG 5). Many report the frustration they feel because they cannot control when the pain appears, its intensity and the moment it forces them to stop or interrupt what they are doing: “*it is very frustrating because you have pain, you want to do something but you can’t, if you try to, it only hurts more, it’s a vicious circle and it affects you even more*” (FG 5). Some participants spoke of the pain as a “waste of time” or “waste of the day”: “*there are days I give up and consider them lost, I have to stay on the sofa and manage the best I can it limits what I can do that day; I have to prepare myself mentally and tell myself that I won’t be able to do anything I had planned that day*” (FG 2).

Similarly, this limitation is intensified during any sports activity that requires any physical exertion (*n* = 19). The greater the intensity or exertion required, the more intense the pain. Some participants (*n* = 6) reported that, even during mild physical exercise, PD appears: “*running or jumping make it hurt more. Making any strong physical effort brings on the pain*”(FG 1);“*I love to practice sports, but because of the pain it’s impossible; it’s incompatible with doing any exercise, no matter how light*” (FG 2). Some students (*n* = 3) report that the experience of pain during sports led them to give up sports: “*living with dysmenorrhea causes me a lot of anxiety, knowing my period is coming, because I practice a lot of sports and the pain makes it totally impossible*” (FG 5); “*when practicing sports it’s very uncomfortable, because you’re doing physical exercise in pain. There are times I avoid doing exercise*.” (FG 1).

Many participants (*n* = 18) described how, apart from the pain that limits their activities, there is also fatigue and lack of energy that the students relate to dysmenorrhea. Students describe it as “*not feeling up to it*”, “*having less energy*”, “*feeling listless*” and tiredness, “*the body becomes a burden*”. This sensation of exhaustion is physical, emotional and mental: “*I feel I have less energy for normal daily things. More despondent about everything. Also, I feel more tired*” (FG 1); “*You feel a tiredness at every level, physical and emotional*.” (FG 3); “*… what happens is I feel very apathetic, totally. The pain brings a lack of appetite for anything and tiredness, a weight in the body that doesn’t let me do anything*…” (FG 5).

### 3.2. Theme 2. Academic Restrictions

Students (*n* = 27) spoke of how PD impacts their class attendance, their learning in class and during practical clinical training, their capacity to concentrate and their academic performance.

Some participants (*n* = 13) acknowledged missing class during episodes of pain, due to the need to rest or feeling discourage: “during that day or days, not only is it the pain, mentally, you don’t feel well I’ve been absent sometimes because the pain was very intense, I’d wake up intending to go but I couldn’t even get dressed.” (FG 5); “the pain is a real limitation, having to sit down for a long time when I’m in class; I try to only go to the compulsory classes, because the pain gets worse if I’m sitting down” (FG 2).

On occasion, students attended class despite believing that it was a waste of time. They referred to being unable to concentrate in class, unable to avoid focussing on the pain, mainly waiting for class to end to return home: “*when I have missed class, it’s because I couldn’t handle the pain. Going to class in pain is to be absent anyway, you are only thinking of how quickly you can get back home*” (FG 5); “*I have also missed class and when I have attended, I’ve had to put up with the pain. I manage the best I can but sometimes I’ve had to leave halfway through.*” (FG 4); “*it really lowers your concentration. You’re not really paying attention. You’re focussed on the pain you’re physically there but your mind is elsewhere concentration is zero. I’ve had to be in class, but I haven’t really attended, I’ve been like part of the decor, thinking only of the pain.*” (FG 5).

In Spain, the study program for a nursing degree requires 2300 h of clinical practice in hospitals or medical centres. Many students (*n* = 18) described how PD hindered their ability to complete their practical training due to tiredness and pain: “*at the hospital you have to run here and there, you have to be on your feet all day. It’s exhausting, and painful. I felt doubly or triply tired*” (FG 2). Students spoke of how clinical practice is more demanding, requiring maximal effort. For this reason, some students (*n* = 3) with PD took medication to be able to continue and keep up their level of activity, so that the pain or tiredness would not become an obstacle: “*during internships I tried do my work but it was difficult, I took some strong pain pills and they more or less worked. But I felt clumsier and more distracted*” (FG 1). The participants spoke of how the pain and tiredness they felt during clinical practice could lead them to make mistakes in hospital tasks: “*with how tired you feel, you try to do things in a quick or organised way so you are able to rest and then the mistakes and errors happen*” (FG 1). All participants agreed that they always try to attend clinical practice, but sometimes they are forced to miss these placements: “*this year, during clinical practice, I was in the surgery and I had to leave because I couldn’t stand the pain I started to feel dizzy because of the pain I felt*” (FG 4).

Primary dysmenorrhea also impacts concentration, time management for study, decreases student performance and the ability to retain knowledge*: “I haven’t had to stop studying, but I can learn three pages in half an hour. When I am in pain, it takes me an hour and a half. I have less concentration*” (FG 2*).* Many students (*n* = 15) reported feeling frustrated, unable to focus on their studies, feeling like trying to study with PD was a waste of time: “*academically, I can perform, I notice it a lot. I have to lie down and put on the electric blanket. Studying is a waste of time*” (FG 1); “*when I try to study, I can’t concentrate because of the pain. I can’t read a single line. So, you leave it and you’ve wasted the whole afternoon because of the pain. I feel frustrated because I’ve lost a whole day without studying*” (FG 2).

### 3.3. Theme 3. Restrictions on Social and Sexual Relations

Many students (*n* = 24) reported that dysmenorrhea conditions their social relations, the ability to meet new people, go out, etc. Over time, participants (*n* = 15) preferred to stay home when suffering from dysmenorrhea, to the point of voluntary social isolation until the pain goes: “*just thinking about having plans and the pain begins, and then I know I have to cancel my plans to go out with friends. It limits my social life*” (FG 2); “*you become totally isolated when you have pain. I don’t want to see anyone, no going to university, no boyfriend, friends, no one. Until the pain goes, I don’t answer my phone and I don’t speak to anyone*” (FG 5).

As a result, many of the students (*n* = 15) described feeling a great deal of anxiety and frustration, facing the uncertainty of having to cancel their plans, of the lack of control: “*I feel a lot of anxiety knowing I will have to cancel everything I had planned that day because I will be feeling bad, I will be in pain. It’s frustrating not being able to control anything*” (FG 5). Consequently, a common strategy adopted among students (*n* = 15) was that they preferred not to make plans, avoiding making plans to meet with friends or isolating themselves in order to avoid cancelling and apologising or having to excuse their absence: “*I prefer not to make plans to avoid cancelling and having to give an explanation and always seeming to have the same excuse of pain*” (FG 4). 

In addition to limiting their social relations, students abandoned their activities or pastimes, making it difficult to maintain these or embark on new ones (*n* = 3): “*I’ve stopped doing things because of the pain. I used to swim, but I had to stop because the week I had my period the pain prevented me from training and competing. I lost my circle of friends*” (FG 2). On the contrary, students (*n* = 12) face this limitation of social relations, admitting that PD is the reason they give up certain activities or the shrinking of their social contacts: “*it doesn’t make me incapable of socialising. If I have a plan and I can’t do it, I admit the reason and that’s it. I don’t feel rejected*” (FG 2).

Regarding sexual relations, some students (*n* = 3) complained of experiencing discomfort or pain during intercourse, reducing the frequency of sexual relations when feeling the pain. Students reported that, on the days of menstruation, this is particularly painful: “*relationships change, I feel under pressure and uncomfortable. It’s not the same. I have less sexual relations but it’s not something that happens constantly*” (FG 2); “*I couldn’t have sexual relations with pain. Nothing. The pain paralyses me completely*” (FG 4); “*when I have my period, during the first days, having sex is unimaginable. Basically, because it hurts*” (FG 2).

## 4. Discussion

The aim of the present study was to describe how nursing students experienced restrictions due to PD. Students with dysmenorrhea described PD as a hindrance affecting three aspects of their lives: daily activities and sports, academic endeavours and studies and their social and sexual relations.

Our findings regarding restrictions on daily activities and sports coincide with the results of Abreu-Sánchez et al. [2], who found that menstrual pain accounted for 62.8% of absenteeism among Spanish female university students and that certain daily activities caused or intensified their menstrual pain. These included siting down, walking and having a full bladder. Additionally, the present study identified feelings of frustration related to the unpredictability of the pain and having to abandon certain activities. Similarly, previous studies have found that irritability, anxiety and symptoms of depression are common among women with dysmenorrhea [8,44]. Possibly, the ineffective management of pain and frustrations due to these limitations may have an emotional impact.

Our results showed that some students give up the practice of sports. In contrast, previous studies [10,45] reported how the practice of certain types of sports can prevent or decrease the intensity of PD. Furthermore, exercise among women with PD has been found to enhance their quality of life [46,47].

This study found that the voluntary abandonment of sports activities was due to the pain caused or worsened by practicing sports. We believe that consulting a health care professional to evaluate alternative treatments could help avoid women having to abandon sports activities. Seeking professional care for the treatment of PD is considered a cost-effective management strategy [48]. However, this is an infrequent solution among young people. 

Regarding academic restrictions, Armour et al. [1] reported in their metanalysis that dysmenorrhea can have a significant academic impact, accounting for 23.5% of absenteeism among university students. Furthermore, 42.4% of those affected experience decreased academic performance and concentration. The participants of the present study reported how they tried to attend clinical practices and avoid being absent despite feeling unwell. This is referred to as “presenteeism”, defined as the practice of going to work despite a medical illness that will prevent full functioning at work [49]. In the field of nursing, presenteeism has been identified as a risk to patient safety due to possible errors in the administration of medication [50,51]. The motives for presenteeism when suffering from menstrual pain may include [2]: students considering menstrual pain as an insufficient reason to be absent, thinking that others do not consider it a valid justification to be absent, anxiety about the negative academic impact of being absent, believing that attending class despite feeling unwell is most responsible, and feelings of guilt when being absent. In Spain, nursing students displayed a 92.7% level of presenteeism due to dysmenorrhea [2]. All of the participants in our study reported attending clinical practices despite feeling pain, having poor concentration and diminished physical stamina. The European Directive 2013/55/UE [52] regulating and requiring 2,300 hours of clinical practice for nursing students may be an important factor in the prevalence of presenteeism among nursing students. 

Concerning social relationships, our participants reported how they voluntarily isolated themselves, gave up hobbies and pastimes and avoided making plans. Many avoided explaining their circumstances for fear this would be taken as a poor excuse or because of the risk of losing friendships [53]. A study by Yagnik et al. [54] pointed to socio-cultural factors for hiding menstruation among women describing several related myths and taboos which inhibit women from speaking freely. Additionally, previous studies [55,56] have found there is a stigma attached to menstrual pain. Even the marketing of women’s health products for menstruation conveys the need to conceal one’s period, despite its psychological nature [57]. Additionally, women avoid sexual relationships when menstruating due to feelings of shame or uncleanliness. This behaviour may be explained by cultural norms or rites which allow or prohibit sexual relations [58]. Furthermore, the presence of dyspareunia [2] is common among women suffering from dysmenorrhea, which justifies the avoidance of sexual relations.

The results of the present study point to the need for women to be given more information from competent health care professionals regarding dysmenorrhea and its management to enhance their wellbeing and empowerment [59]. A future line of research may be to analyse how nursing students deal with dysmenorrhea focusing on their pain management strategies. The authors, in line with Armour et al. [60], consider that further research is needed to investigate the impact of different pain management strategies on women’s lives. Methodologically rigorous randomized clinical trials could demonstrate the effects of the same on the intensity of menstrual pain and its impact on women’s daily lives.

Additionally, it is essential to develop strategies to provide information and raise awareness (health education) on dysmenorrhea aimed at the entire community in order to dismantle myths and destigmatise menstruation [53,54,61]. These strategies should be designed based on an analysis of students’ needs [62] and be provided from early school years through to higher education [63]. Alimoradi et al. highlight the need for the involvement and coordination of political leaders, healthcare professionals and educators in sexual and reproductive education policies [59]. Furthermore, specific actions aimed at addressing the needs of nursing students should include a review of absenteeism regulations for clinical practice and internships when related to menstrual problems, adjusting the system for the recovery of missed classes and thus avoiding the phenomenon of presenteeism and the risk of treatment errors during internships in medical centres and hospitals [2].

Finally, the main strength of this study is that it provides a description of how young women live with the restrictions imposed by dysmenorrhea, revealing first-hand accounts that can influence patient care. To the best of our knowledge, this is one of the few studies that describes what it is like for young female nursing students to live with PD.

This study has limitations. Firstly, the results can’t be extrapolated. Secondly, within the FGs, the participants may tend to unify their criteria, and avoid comparison and dialogue, which may produce uniform results [30]. To control this effect, the moderator asked the participants questions to determine their individual perspective and to counteract this tendency. In conducting focus groups via the Zoom digital platform, the study authors applied a detailed data collection procedure, using a rotational system to raise hands and speak (see FG procedure)**.** Finally, as there is currently no specific and conclusive test for the differential diagnosis of primary and secondary dysmenorrhea, some participants may have had secondary dysmenorrhea although without identifying it as such. However, this limitation is found in most studies on primary dysmenorrhea [2,7,8,64,65], furthermore, in the current study this was minimized, considering the recommendations of the Primary Dysmenorrhea Consensus Guideline of the Society of Obstetricians and Gynaecologists of Canada and the Committee Opinion on Adolescent Health Care Dysmenorrhea and Endometriosis in the Adolescent developed by the American College of Obstetricians and Gynecologists [5,39]. Thus, we avoided including women for whom, according to these recommendations, additional testing was recommended because of suspected secondary dysmenorrhea or other pathologies with pelvic pain.

## 5. Conclusions

This group of nursing students with PD experienced restrictions on their daily activities and sports pastimes, their academic studies and their social and sexual relationships. Considering the prevalence of this problem, these results highlight the need for effective management of primary dysmenorrhea, with the involvement of health care professionals and the community at large.

## Figures and Tables

**Table 1 ijerph-17-08527-t001:** Data collection information.

FG	FG Session 1: Duration (minutes)	FG Session 2: Duration (minutes)	Total duration FG Sessions (minutes)	Participants	Age, Mean
FG1	51	52	103	6	21.5
FG2	65	46	111	6	22
FG3	51	47	98	6	24
FG4	45	45	90	6	24
FG5	78	42	120	9	22.3

FG: Focus Group.

**Table 2 ijerph-17-08527-t002:** Semi-structured focus group question guide.

Research Area	Questions
Living with the pain of dysmenorrhea	What is most relevant to you about this pain? Could you explain your pain?
Repercussions of pain: Daily activities	How does this pain affect you? What is most relevant regarding living with pain?
Repercussions of pain: Nursing studies	Does the pain have any impact on academic performance? What is most relevant about the pain regarding your studies?
Repercussions of pain: Relationships and family	Has the pain influenced your relationships? How about with your partner and the rest of the family?

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
