# Peer review of "Living with Restrictions. The Perspective of Nursing Students with Primary Dysmenorrhea"

_ijerph, 2020, doi:10.3390/ijerph17228527_

Round 1
Reviewer 1 Report
Thank you for your recent submission to the IJERPH. The topic of your paper is very interesting, and might be of interest to our readers. You have also presented an interesting framework. I have a few suggestions:
1 - Insert the main limitations in the article
2- Insert suggestions for future research
3 - Present the practical, toric, and political implications of the study.
Author Response
ID del manuscrito : ijerph-975554
Título: Vivir con restricciones. La perspectiva de los estudiantes de enfermería con dismenorrea primaria
Nos gustaría agradecer a los editores y revisores por su cuidadosa consideración de nuestro manuscrito. También nos gustaría agradecer las sugerencias de los revisores, que creemos que han mejorado la calidad del manuscrito. Hemos destacado todos los cambios que hemos realizado a lo largo del texto. A continuación, encontrará una lista detallada de cómo hemos abordado cada comentario.
REVISOR 1.
Gracias por su reciente presentación al IJERPH. El tema de su artículo es muy interesante y puede ser de interés para nuestros lectores. También ha presentado un marco interesante, tengo algunas sugerencias:
1 - Insert the main limitations in the article
Response: We had followed the reviewer´s suggestions. We have included the following in the discussion section.
This study has limitations. Firstly, the results cannot be extrapolated. Secondly, within the FGs, the participants may tend to unify their criteria, and avoid comparison and dialogue, which may produce uniform results [30]. To control this effect, the moderator asked the participants questions to determine their individual perspective and to counteract this tendency. In conducting focus groups via the Zoom digital platform, the study authors applied a detailed data collection procedure, using a rotational system to raise hands and speak (see FG procedure).
2- Insert suggestions for future research
Response: We had followed the reviewer´s suggestions. We have included the following in the discussion section.
A future line of research may be to analyse how nursing students deal with dysmenorrhea focusing on their pain management strategies. The authors, in line with Armour et al. [60], consider that further research is needed to investigate the impact of different pain management strategies on women's lives. Methodologically rigorous randomized clinical trials could demonstrate the effects of the same on the intensity of menstrual pain and its impact on women's daily lives.
Reference
Armour, M.; Smith, C.A.; Steel, K.A.; MacMillan, F. The effectiveness of self-care and lifestyle interventions in primary dysmenorrhea: A systematic review and meta-analysis. BMC Complement. Altern. Med. 2019, 19.
3 - Present the practical, topic, and political implications of the study.
Response: We have followed the reviewer´s suggestions. This information is included in the text. See discussion section below:
Additionally, it is essential to develop strategies to provide information and raise awareness (health education) on dysmenorrhea aimed at the entire community in order to dismantle myths and destigmatise menstruation [53,54,61]. These strategies should be designed based on an analysis of students’ needs [62] and be provided from early school years through to higher education [63]. Alimoradi et al. highlight the need for the involvement and coordination of political leaders, healthcare professionals and educators in sexual and reproductive education policies [59]. Furthermore, specific actions aimed at addressing the needs of nursing students should include a review of absenteeism regulations for clinical practice and internships when related to menstrual problems, adjusting the system for the recovery of missed classes and thus avoiding the phenomenon of presenteeism and the risk of treatment errors during internships in medical centres and hospitals [2].
Esperamos que esté satisfecho con la revisión y que el manuscrito esté ahora apto para su publicación en IJERPH.
Sinceramente,
Los autores

Reviewer 2 Report
I think that Dysmenorrhea is really an important problem causing too much pain for women and it must be studied.
The title should include Primary Dysmenorrhea, since all the girls in the study presented this type of Dysmenorrhea, and the secondary was an exclusion criteria.
In the abstract, a verb is missing in the phrase: A video meeting platform to conduct the focus groups. Include was used to conduct...
It consists of a qualitative study but it should include how many cited restrictions in the activities and how many gave up the practice of sports or missed classes. This could emphasize that it is a problem affecting many women.
It was also missing any question about self-medication, which was very relevant.
I am not familiar with the term scholarship in the phrase: Scholarship distinguishes between two types of dysmenorrhea.
On page 3, line 96, please write FG as Focus Group (FG).
It would be nice to include a table with the cited phrases in the Results section, but I understand that the authors preferred to write in the text.
Author Response
Manuscript ID: ijerph-975554
Title: Living with restrictions. The perspective of nursing students with Primary Dysmenorrhea
We would like to thank the Editors and the Reviewers for their careful consideration of our manuscript. We would also like to thank the Reviewers’ suggestions, which we believe have enhanced the quality of the manuscript. We have highlighted all the changes we have made throughout the text. Below please find a detailed list of how we have addressed each comment.
REVIEWER 2
I think that Dysmenorrhea is really an important problem causing too much pain for women and it must be studied. The title should include Primary Dysmenorrhea, since all the girls in the study presented this type of Dysmenorrhea, and the secondary was an exclusion criteria.
Response: We had followed the reviewer´s suggestions and the title has been edited to include primary dysmenorrhea.
In the abstract, a verb is missing in the phrase: A video meeting platform to conduct the focus groups. Include was used to conduct...
Response: We have followed the reviewer´s suggestions and edited the abstract.
It consists of a qualitative study but it should include how many cited restrictions in the activities and how many gave up the practice of sports or missed classes. This could emphasize that it is a problem affecting many women.
Response: We have followed the reviewer´s suggestions. We have included this information in the results section.
It was also missing any question about self-medication, which was very relevant.
Response: We agree with reviewer. However, although a specific question was not included in the question guide, the issue of medication consumption, and self-medication emerged in the focus groups. In the event that a relevant area related to the topic at hand arose, the moderator began to ask the participants about it. In the present work, the focus was on describing the restrictions of dysmenorrhea, not the effect or consumption of self-medication for pain control.
We included the following in the data collection section:
Nonetheless, data collection in qualitative studies is flexible, allowing the researcher to ask open-ended questions on topics that are relevant to the participants in relation to the research question [26]. Consequently, during the FGs, the moderator asked about those areas of interest that the participants raised in relation to the research question.
I am not familiar with the term scholarship in the phrase: Scholarship distinguishes between two types of dysmenorrhea.
Response: We agree with reviewer. We have rewritten this sentence. We have included the following in the introduction section:
“Two types of dysmenorrhea are distinguished in the literature…”
We have included these new references:
- Proctor, M.; Farquhar, C. Diagnosis and management of dysmenorrhoea. Br. Med. J. 2006, 332, 1134–1138.
- Burnett, M.; Lemyre, M. Primary Dysmenorrhea Consensus Guideline. J. Obstet. Gynaecol. Canada 2017, 39, 585–595.
On page 3, line 96, please write FG as Focus Group (FG).
Response: Previously, in the design section, the meaning of the FG acronym is explained to readers.
- Materials and Methods
2.1. Design
A qualitative exploratory study was conducted using focus groups (FGs) [24].
It would be nice to include a table with the cited phrases in the Results section, but I understand that the authors preferred to write in the text.
Response: Thank you for your suggestion. On this point the authors believe that integrating the narratives and quotes within the narrative would improve the flow of the paper and emphasize findings in the words of the participants themselves, rather than using a separate table.
We hope that you are satisfied with the revision and that the manuscript is now suitable for publication in IJERPH.
Sincerely,
The Authors

Reviewer 3 Report
Dear Authors,
The aim of this qualitative exploratory study was to evaluate how nursing students experienced restrictions resulting from primary dysmenorrhea.You examined 33 nursing students with primary dysmenorrhea. Data was collected from five focus groups (two sessions each) and the field notes of 10 researchers. A video meeting platform was used for the focus groups. This group of nursing students experienced restrictions on their daily activities and sports pastimes, their academic studies and their social and sexual relationships. You concluded that primary dysmenorrhea caused restrictions in the personal, social and academic life of the nursing students. In the last years, primary dysmenorrhea has received great attention and many papers have been published. Although the topic of the study is captivating and of great academic interest, the manuscript does not reach the standard to be published in the IJERPH in the present form.
Comments:
- The inclusion and exclusion criteria should be described in more detail.
- Is not described how the primary dysmenorrhea was evaluated.
- Is not described how underlying causes of dysmenorrhea were excluded.
- Is not described socio-demographic characteristics of participants in the study.
- Is not described whether the patients took analgesics or other drugs and how these changed the impact of dysmenorrhea on their activities.
- There are many validated questionnaires on quality of life in patients with primary dysmenorrhea, that can estimated, with a score, the quality of life. Using the focus groups, the participants may tend to unify their criteria, and avoid comparison and dialogue. This may produce uniform results.
- As said in the manuscript, the results cannot be extrapolated.
Author Response
Manuscript ID: ijerph-975554
Title: Living with restrictions. The perspective of nursing students with Primary Dysmenorrhea
We would like to thank the Editors and the Reviewers for their careful consideration of our manuscript. We would also like to thank the Reviewers’ suggestions, which we believe have enhanced the quality of the manuscript. We have highlighted all the changes we have made throughout the text. Below please find a detailed list of how we have addressed each comment.
REVIEWER 3.
The aim of this qualitative exploratory study was to evaluate how nursing students experienced restrictions resulting from primary dysmenorrhea.You examined 33 nursing students with primary dysmenorrhea. Data was collected from five focus groups (two sessions each) and the field notes of 10 researchers. A video meeting platform was used for the focus groups. This group of nursing students experienced restrictions on their daily activities and sports pastimes, their academic studies and their social and sexual relationships. You concluded that primary dysmenorrhea caused restrictions in the personal, social and academic life of the nursing students. In the last years, primary dysmenorrhea has received great attention and many papers have been published. Although the topic of the study is captivating and of great academic interest, the manuscript does not reach the standard to be published in the IJERPH in the present form.
Comments:
The inclusion and exclusion criteria should be described in more detail. Is not described how the primary dysmenorrhea was evaluated. Is not described how underlying causes of dysmenorrhea were excluded.
Response: The inclusion and exlcusion criteria have been described in more detail in the new version of the manuscript.
The inclusion criteria consisted of: a) nursing students enrolled in the University of Huelva (https://enfe.acentoweb.com/) during the study period, b) PD: who acknowledged experiencing menstrual pain for which no underlying cause had been diagnosed [7, 31], c) suffering from menstrual pain at least once in the last six months, with at least three periods per year [7,18,19], d) moderate-severe pain intensity based on the visual analogue scale (VAS equal to or greater than 4 out of 10 [8,32,33], e) for at least the three cycles prior to the study [34,35], f) with normal menstrual characteristics in terms of cycle length, duration of menstruation, quantity and regularity. Normal menstrual characteristics were considered as periods occurring every 24 to 38 days, on a regular basis, with bleeding that lasts 4.5 to 8 days, and 5 to 80 mL blood loss per cycle [36-38] and g) not being a candidate for requesting diagnostic tests due to suspicion of suffering secondary dysmenorrhea or any other diagnosed pelvic or gynaecological problem based on the recommendations of the Primary Dysmenorrhea Consensus Guideline of the Society of Obstetricians and Gynaecologists of Canada and the Committee Opinion on Adolescent Health Care Dysmenorrhea and Endometriosis in the Adolescent developed by the American College of Obstetricians and Gynecologists [5,34,39].
The exclusion criteria consisted of: being diagnosed with secondary dysmenorrhea (endometriosis, adenomyosis, uterine myomas, cervical stenosis and obstructive lesions of the genital tract) or any other diagnosed pelvic or gynaecological problem (pelvic inflammatory disease, pelvic adhesions, irritable bowel syndrome, inflammatory bowel disease, interstitial cystitis, mood disorders and myofascial pain) or menstrual characteristics that may raise suspicion of these pathologies [5,39], and unwillingness to participate in the study. Thirty-three female nursing students participated in the study. There were no dropouts.
We included new references:
- Onieva-Zafra, M.D.; Fernández-Martínez, E.; Abreu-Sánchez, A.; Iglesias-López, M.T.; García-Padilla, F.M.; Pedregal-González, M.; Parra-Fernández, M.L. Relationship between diet, menstrual pain and other menstrual characteristics among Spanish students. Nutrients2020, 12, 1759.
- Schoep, M.E.; Nieboer, T.E.; van der Zanden, M.; Braat, D.D.M.; Nap, A.W. The impact of menstrual symptoms on everyday life: A survey among 42,879 women. Am. J. Obstet. Gynecol. 2019, 220, 569.e1–569.e7Yang, M.; Chen, X.; Bo, L.; Lao, L.; Chen, J.; Yu, S.; Yu, Z.; Tang, H.; Yi, L.; Wu, X.; et al. Moxibustion for pain relief in patients with primary dysmenorrhea: A randomized controlled trial. PLoS One 2017, 12
- Gómez-Escalonilla Lorenzo, B.; Rodríguez Guardia, Á.; Marroyo Gordo, J.M.; de las Mozas Lillo, R. Frecuencia y características de la dismenorrea en mujeres de la zona de salud de Torrijos (Toledo). Clin.2010, 20, 32–35Mihm, M.; Gangooly, S.; Muttukrishna, S. The normal menstrual cycle in women. Anim. Reprod. Sci. 2011, 124, 229–236.
- Munro, M.G. Classification of menstrual bleeding disorders. Rev. Endocr. Metab. Disord. 2012, 13, 225–234.
- Abreu-Sánchez, A.; Parra-Fernández, M.L.; Onieva-Zafra, M.D.; Fernández-Martínez, E. Perception of menstrual normality and abnormality in spanish female nursing students. Int. J. Environ. Res. Public Health 2020, 17, 1–12.
- Burnett, M.; Lemyre, M. Primary Dysmenorrhea Consensus Guideline. J. Obstet. Gynaecol. Canada 2017, 39, 585–595.
- Geri D. Hewitt, M.; Karen R. Gerancher, M. ACOG COMMITTEE OPINION Number 760 Committee on Adolescent Health Care Dysmenorrhea and Endometriosis in the Adolescent; Whashington, 2018.
- Yang, M.; Chen, X.; Bo, L.; Lao, L.; Chen, J.; Yu, S.; Yu, Z.; Tang, H.; Yi, L.; Wu, X.; et al. Moxibustion for pain relief in patients with primary dysmenorrhea: A randomized controlled trial. PLoS One 2017, 12.
No diagnostic test or pelvic examination was performed on the participants to determine the type of dysmenorrhea they were experiencing or to rule out other pathology since no specific and conclusive test exists for this purpose, and, according to the Dysmenorrhea Consensus Guideline of the Society of Obstetricians and Gynaecologists of Canada and Committee on Adolescent Health Care developed by the American College of Obstetricians and Gynecologists (Geri et al., 2018; Burnett & Lemyre, 2017) the performance of any diagnostic tests or pelvic examinations in women with menstrual pain who, based on gynecological and menstrual data, do not suspect any associated pathology is considered unnecessary.
References:
- Burnett, M.; Lemyre, M. Primary Dysmenorrhea Consensus Guideline. Obstet. Gynaecol. Canada 2017, 39, 585–595.
- Geri D. Hewitt, M.; Karen R. Gerancher, M. ACOG COMMITTEE OPINION Number 760 Committee on Adolescent Health Care Dysmenorrhea and Endometriosis in the Adolescent; Whashington, 2018.
The authors believe that it is necessary to comment on this in the limitations:
Finally, as there is currently no specific and conclusive test for the differential diagnosis of primary and secondary dysmenorrhea, some participants may have had secondary dysmenorrhea although without identifying it as such. However, this limitation is found in most studies on primary dysmenorrhea [2,7,8,64,65], furthermore, in the current study this was minimized, considering the recommendations of the Primary Dysmenorrhea Consensus Guideline of the Society of Obstetricians and Gynaecologists of Canada and the Committee Opinion on Adolescent Health Care Dysmenorrhea and Endometriosis in the Adolescent developed by the American College of Obstetricians and Gynecologists [5,39]. Thus, we avoided including women for whom, according to these recommendations, additional testing was recommended because of suspected secondary dysmenorrhea or other pathologies with pelvic pain.
Is not described socio-demographic characteristics of participants in the study.
Response: In relation to your comment, further socio-demographic information and other characteristics of interest regarding the participants have been added in the first paragraph of the results section:
Thirty-three female nursing students participated in the study. The mean age of the participants was 22.72 (SD 3.46) years and the BMI was 21.52 (SD 2.50)kg/m2. Up to 87.9% lived in an urban setting compared to 12.1% in a rural setting. A total of 84.8% regularly took some type of medication to manage their menstrual pain, whereas 81.8% used non-steroidal anti-inflammatory drugs and 21.2% used hormonal contraceptives.
Is not described whether the patients took analgesics or other drugs and how these changed the impact of dysmenorrhea on their activities.
Response: Information on consumption of analgesics and other drugs obtained in the initial form on characteristics of the sample has been included in the results section together with the socio-demographic data:” …El 84.8% tomaba habitualmente algún tipo de fármaco para manejar su dolor menstrual. El 81.8% usaba antiinflamatorios no esteroideos y el 21.2% consumía anticonceptivos hormonales.”
The impact of analgesic or other drug use on dysmenorrhea and its activities was not analyzed because it deviated from the objective of this study.
There are many validated questionnaires on quality of life in patients with primary dysmenorrhea, that can estimated, with a score, the quality of life.
Response: The authors agree that there are many questionnaires that measure quality of life. However, in the present study the research question was not to study quality of life. For this reason, no questionnaires measuring quality of life have been used.
It is true that certain qualitative designs (ethnography or qualitative case-studies) (Curry et al., 2009; Creswell & Poth, 2018; Carpenter & Suto, 2008) or mixed methods research (Curry & Nuñez-Smith, 2015), which, besides the use of qualitative data (narrative, visual) from data collection tools such as in depth interviews, focus groups and observation, also use other tools such as questionnaires, scales, etc. (Creswell & Poth, 2018; Carpenter & Suto, 2008). However, this is not the case in this study. Here we used a qualitative exploratory study design, characterized by gathering the perspective of participants, via focus groups (Curry et al., 2009).
“Qualitative research may be conducted prior to quantitative research to set the direction for exploration with quantitative methods, or as follow-up to quantitative studies, where it can aid in interpretation (…) Qualitative research can also help identify which aspects of the intervention are valued, or not, and why (…) So it can be useful to read both a study evaluating the effects of an intervention and a complementary study exploring participants’ experiences of the intervention (…) There are other areas of qualitative research that are highly relevant to practice. Studies that have as their objective to understand clients’ health-related perceptions and explore patients’ experiences with therapy can be very useful.”(Herbert et al., 2011.p.28-29).
References:
- Carpenter C, Suto M. Qualitative research for occupational and physical therapists: A practical guide. Oxford: Black-Well Publishing, 2008.
- Creswell JW, Poth CN. Qualitative inquiry and research design. Choosing among five approaches. 4 ed. Thousand Oaks: SAGE, 2018.
- Curry LA, Nembhard IM, Bradley EH. Qualitative and Mixed Methods Provide Unique Contributions to Outcomes Research. Circulation. 2009; 119: 1442–1452.
- Curry L, Nuñez-Smith M. Mixed methods in health sciences research. Thousand Oaks, CA: Sage publications; 2015.
- Herbert R, Jamtvedt G, Hagen KB, Mead J. Practical Evidence-Based Physiotherapy. 2 ed. Elsevier Churchill Livingstone: Oxford, England, 2011.p. 28-29.
Using the focus groups, the participants may tend to unify their criteria, and avoid comparison and dialogue. This may produce uniform results.
Response: We agree with the reviewer´s suggestions. On the other hand, the authors are aware that the Focus Groups, if not conducted correctly, can provoke a uniform dialogue. We believe that it is necessary to point out this aspect in the discussion and to describe how the process has been controlled and how the tendency to unify the perspective of the participants has been avoided.
For this reason, we described the following in the limitations section:
Secondly, within the FGs, the participants may tend to unify their criteria, and avoid comparison and dialogue, which may produce uniform results [30]. To control this effect, the moderator asked the participants questions to determine their individual perspective and to counteract this tendency.
As said in the manuscript, the results cannot be extrapolated.
Response: We agree with the reviewer´s suggestion.
However, qualitative research is a type of research that is based on studying the experience and perspective of people in certain situations of health and illness and how they experience the impact of interventions and health technologies (Curry et al., 2009; Creswell & Poth, 2018; Carpenter & Suto, 2008). In the PubMed database, in their Medical Subject Heading (MeSH) section, “qualitative research” is defined as: “Any type of research that employs nonnumeric information to explore individual or group characteristics, producing findings not arrived at by statistical procedures or other quantitative means.”
Creswell & Poth (2018) defined qualitative research as: “Qualitative research begins with assumptions and the use of interpretive/theoretical frameworks that inform the study of research problems addressing the meaning individuals or groups ascribe to a social or human problem. To study this problem, qualitative researchers use an emerging qualitative approach to inquiry, the collection of data in a natural setting sensitive to the people and places under study, and data analysis that is both inductive and deductive and establishes patterns or themes. The final written report or presentation includes the voices of participants, the reflexivity of the researcher, a complex description and interpretation of the problem, and its contribution to the literature or a call for change.”(p.8)
Also, in qualitative research there is not a number of participants that one can calculate previously (Creswell & Poth, 2018; Carpenter & Suto, 2008), in the same manner, in qualitative research we do not seek to extrapolate the results to the general population (Creswell & Poth, 2018), rather, the attempt is to deepen our knowledge of the impact of programs and interventions on groups of people who suffer illnesses or specific situations of health or illness.
“The intent in qualitative research is not to generalize the information but to elucidate the particular, the specific.” (Creswell & Poth, 2018.p.158).
“In quantitative studies, power calculations determines which sample size (N) is necessary to demonstrate effects of a certain magnitude from an intervention. For qualitative interview studies, no similar standards for assessment of sample size exist.” (Malterud et al., 2015.p.1)
References:
- Carpenter C, Suto M. Qualitative research for occupational and physical therapists: A practical guide. Oxford: Black-Well Publishing, 2008.
- Creswell JW, Poth CN. Qualitative inquiry and research design. Choosing among five approaches. 4 ed. Thousand Oaks: SAGE, 2018.
- Curry LA, Nembhard IM, Bradley EH. Qualitative and Mixed Methods Provide Unique Contributions to Outcomes Research. Circulation. 2009; 119: 1442–1452.
- Malterud K, Siersma VD, Guassora AD. Sample Size in Qualitative Interview Studies: Guided by Information Power. Qual Health Res. 2015 Nov 27. DOI: 10.1177/1049732315617444
We hope that you are satisfied with the revision and that the manuscript is now suitable for publication in IJERPH.
Sincerely,
The Authors

Round 2
Reviewer 3 Report
Dear Authors, thank you for your extended revision of the manuscript. Now, I think the article is ready for publication on IJERPH